# Epidemiological Characteristics of Varicella under Different Immunisation Strategies in Suzhou Prefecture, Jiangsu Province

**DOI:** 10.3390/vaccines10101745

**Published:** 2022-10-19

**Authors:** Zhuoyu Zhang, Na Liu, Jun Zhang, Juan Xu, Wenyu Wang, Jiaqi Xiao, Tianyu Wang, Lin Luan, Yunyan Zhang

**Affiliations:** 1School of Public Health, Nanjing Medical University, Nanjing 211166, China; 2Suzhou Center for Disease Control and Prevention, Suzhou 215000, China; 3School of Public Health, Medical College of Soochow University, Suzhou 215000, China

**Keywords:** varicella, vaccine coverage, immunisation strategy

## Abstract

Background: The varicella vaccine is excluded from the Chinese national immunisation programme but is included in the local expanded programme on immunisation (EPI) in the Suzhou Prefecture. This study investigated the epidemiological characteristics of the varicella cases during the implementation of different immunisation strategies in the Suzhou Prefecture, Jiangsu Province. Methods: In this study, we used descriptive statistics. Information on reported instances from 2012 to 2021 was first retrieved. Data on varicella cases were collected from the China Information System for Disease Control and Prevention (CISDCP). Similarly, information on vaccinated children was obtained from the Jiangsu Province Vaccination Integrated Service Management Information System (JPVISMIS). The census data in this study was procured from the Suzhou Bureau of Statistics. Results: From 2012 to 2021, a total of 118,031 cases of varicella were reported in Suzhou, and the average annual reported incidence was 91.35 per 100,000. The average yearly incidence after implementing the two-dose varicella vaccination decreased by 41.57% compared with the implementation of one dose. This study demonstrates two annual incidence peaks, a small peak between April and July and a prominent peak between October and January. It is also possible that this seasonal distribution is related to the geography of Suzhou. The average annual reported incidence between districts with a statistically significant difference (χ^2^ = 98.077, *p* < 0.05). The one-dose varicella vaccination coverage gradually increased from 55.34% in 2012 to 89.06% in 2021 and the two-dose varicella vaccination coverage gradually increased from 0.27% in 2012 to 82.17% in 2021. Conclusions: Administering the varicella vaccine in the local EPI has significantly decreased the incidence rate and the total number of cases. A two-dose vaccination schedule is still the best vaccination strategy for varicella vaccine effectiveness.

## 1. Introduction

Varicella is a childhood disease caused by a varicella-zoster virus (VZV) infection. It primarily affects the unvaccinated population [1]. The disease is transmitted via respiratory droplets [2], direct contact with varicella herpes fluid, and contact with contaminated utensils. This condition is commonly observed in collective units such as childcare facilities and schools [3], and the age of onset in children is generally two to eight years [1,4]. In addition, varicella has a seasonal onset, with the peak incidence occurring in spring and winter [5,6], during which most children are in school for group study and prone to outbreaks. Preventing and controlling varicella is now increasingly prioritised by infectious disease control and school health.

The average annual incidence of varicella in China between 2016 and 2019 was 55.05 per 100,000 [7], which was an increasing trend compared with the average yearly incidence between 2005 and 2015 (23.04 per 100,000) [8]. The incidence of varicella in the Jiangsu Province from 2016 to 2019 illustrated a year-on-year increase, with an average annual reported incidence being the highest in China, at 117.17 per 100,000 [7].

The varicella vaccine is an effective preventive measure against varicella disease. It was granted a general licence in the USA in 1995 [9], China did not start vaccination against varicella until 2000. The varicella vaccine was included in Suzhou’s local EPI on 1 April 2018 to further manage the varicella epidemic.

This study aims to describe the epidemiological changes caused by varicella before and after implementing the updated vaccination schedule in Suzhou and elucidate the association between varicella’s morbidity characteristics and varicella vaccination.

## 2. Materials and Methods

### 2.1. Setting

Suzhou is located in Jiangsu Province, China, and has a population of 12.75 million in 10 districts and counties, with approximately 200 vaccination clinics for children. On 1 June 2017, Suzhou was the first city in China to include varicella in passive surveillance. The one-dose varicella vaccine (VarV-1) regime was introduced into the local EPI on 1 April 2018, and the two-dose varicella vaccine (VarV-2) was included in the EPI on 1 January 2020.

### 2.2. Data Resources and Target Population

All varicella cases are documented in the China Information System for Disease Control and Prevention (CISDCP). Varicella cases are classified as clinically diagnosed, confirmed, or suspected. Varicella cases are recorded by the onset date, and the statistical time frame was selected from 1 January 2012 to 31 December 2021. The case information was retrieved from the CISDCP on 30 June 2022. It included the patient’s name, sex, date of birth, date of onset, age, and address. The census data using the total population of Suzhou were obtained from the seventh census.

The population studied for the varicella vaccination coverage in Suzhou from 2012 to 2021 was children aged one to six in the Jiangsu Province Vaccination Integrated Service Management Information System (JPVISMIS) each year. Vaccination coverage data were obtained from JPVISMIS on 28 September 2022.

### 2.3. Estimation of Varicella Vaccination Coverage

In the local Suzhou EPI, VarV-1 was recommended for children aged ≥ 12 months, and VarV-2 was recommended for children aged ≥ 48 months and at least 3 months after the first dose. Children born after 1 January 2016 in Suzhou are eligible for the above policies.

The denominator is the children from the vaccination system in JPVISMIS, and the numerator is the others who have completed the vaccination. Accordingly, the varicella vaccine coverage in Suzhou from 2012 to 2021 was determined.

To evaluate the impact of varicella vaccination, the years 2012–2021 were classified into three time periods: 2012–2017, the period before the government implemented free varicella vaccination; 2018–2019, the period of the one-dose vaccination schedule; and 2020–2021, the period of the two-dose vaccination schedule.

### 2.4. Descriptive Statistical Analysis

Total and age-specific incidence rates were calculated based on the number of varicella cases and the study population from 1 January 2012 to 31 December 2021. The information on varicella cases was retrieved from the CISDCP, and the number of cases in each area of Suzhou was recorded according to the address information therein. The incidence rate of each area was then calculated.

Microsoft Office Excel (version 2016) and R for Windows (version 4.1.3) were used for data processing and statistical analyses. Mapping of districts used ArcGIS (version 10.8). Descriptive statistics were applied to describe the incidence in terms of person, place and time. Chi-squared or Pearson chi-squared tests were used to compare the incidences rates and vaccine coverage. The main focus was on changes in morbidity and vaccine coverage before and after the inclusion of the varicella vaccine in EPI and before and after the inclusion of VarV-1 and VarV-2. The level of statistical significance was set at *p* < 0.05.

## 3. Results

### 3.1. Demographic Characteristics and Age Distribution

From 2012 to 2021, a total of 118,031 cases of varicella disease were documented in Suzhou, and the average annual reported incidence was 91.35 per 100,000. The greatest reported incidence of varicella occurred in 2012 with a rate of 49.17 per 100,000, and increased to 207.98 per 100,000 in 2019, and then decreased to 101.26 per 100,000 in 2021 (Table 1). The varicella incidence in 2018–2019 had a statistically significant with that in 2012–2017 and 2020–2021, which is caused by the inclusion of the varicella vaccine in the EPI in Suzhou in 2018.

During 2012–2021, 18.00%, 40.11%, 13.56%, 12.06%, and 16.28% of cases were reported in ≤4, 5–9, 10–14, 15–19, and ≥20 years, respectively. Of all patients, 58.11% of cases were belong to the group of aged <9 years and were concentrated in kindergarten and primary school. Figure 1a shows the age distribution of varicella incidences in Suzhou. The curve shows a sudden increase at aged 2 years, and the peak occurred at aged 6 years, followed by a gradual decrease in incidence. The age-specific incidences of varicella in children aged 4–9 years were the highest. The incidence rate decreases rapidly in the group of aged 9–14 years and then tapers off in following groups (≥15 years).

Figure 1b shows a fluctuation in the incidence of varicella in children within one year of the varicella vaccination in Suzhou. The incidence rate increased from 49.17 per 100,000 before implementation to 207.98 per 100,000 after implementation (χ^2^ = 98.077, *p* < 0.05). A large upward trend was noticed in the incidence of varicella in children starting at 2.5 years, and a decreasing trend from year to year at the aged 6 years. This finding agrees with the results obtained in Figure 1a. The male-to-female ratio with varicella was 1.15. The average annual incidence for males and females was 44.61 per 100,000 and 39.06 per 100,000, respectively (χ^2^ = 1074.9, *p* < 0.05).

### 3.2. Geographic Distribution

Table 2 shows the average annual incidence rates for each district in Suzhou from 2012 to 2021. The differences in the average annual incidence rates between the districts were statistically significant (χ^2^ = 85.482, *p* < 0.05). Table 2 shows that the differences in incidence in Suzhou were statistically significant before and after the inclusion of the varicella vaccine in EPI and compared to the incidence by region in 2018–2019 and 2020–2021.

Figure 2 illustrates that the colour deepened in some areas in 2017 compared to 2012–2016. Though the colour showed a continuous process of deepening in all the area of Suzhou in 2018 and 2019, the colour became lighter again in 2020–2021.

From 2012 to 2021, the top three average annual incidences in Suzhou occurred in Suzhou Industrial Park (SIP) (118.28/100,000), Changshu (113.29/100,000) and Suzhou national Hi-tech District (SND) (117.01/100,000). The highest incidences in 2012, 2013, 2014–2017, and 2018–2021 occurred in Wuzhong (69.95/100,000), SIP (63.05/100,000), SND (150.10/100,000), and Changshu (229.81/100,000), in that order.

After implementing two-dose vaccination, the number of cases in each district decreased by approximately 46.94% compared with the one-dose vaccination schedule. The top three districts and cities with a notable decrease were SIP (69.64%), Gusu (60.51%) and SNP (59.59%).

### 3.3. Seasonal Distribution

Varicella cases were reported in all months between 2012 and 2021, with two annual peaks of varicella incidence, a small peak from April to July and a prominent peak from October to January (Figure 3).

The average monthly incidence before the varicella vaccine implementation was 3.04 per 100,000, which was affected by local policies increasing the average monthly incidence to 17.33 per 100,000 in 2018–2019 and decreasing to 8.44 per 100,000 after the VarV-2 implementation. The highest annual average varicella incidence months in 2012–2017, 2018–2019, and 2020–2021 were all in December, and their incidences were 8.80/100,000, 48.94/100,000, and 18.23/100,000, respectively.

### 3.4. Estimation of Varicella Vaccination Coverage

Figure 4a,b depicts the relationship between varicella vaccination coverage and incidence rate or median age in Suzhou during 2012–2021.VarV-1 vaccination coverage was lower; however, it increased considerably to 89.06% in 2021. This percentage remained unchanged between 49.10% and 59.57% for nearly 6 years from 2012 to 2017. From 2012 to 2017, the VarV-1 vaccination coverage was at about 55%. As a result, the Suzhou government started free VarV-1 vaccination on 1 April 2018 to meet the great demand for this vaccine and increase the vaccination coverage achieve a certain level to provide protect to the population. The VarV-2 vaccination coverage witnessed a steady growth from 0.27% in 2012, which remained at this level until 2019. It sharply increased to 82.17% in 2021.

Figure 4a shows the incidence of varicella by year in Suzhou from 2012 to 2021. A slight fluctuation in the incidence of varicella was observed from 2012 to 2014. This change may be attributed to other factors, thus making health workers more motivated to report varicella. Following the inclusion of varicella in passive surveillance on 1 June 2017, varicella incidence rose sharply from 2017 to 2018, suggesting an underestimation of previous varicella incidence. From 2012 to 2017, VarV-1 did not show a notable downward trend in varicella incidence; however, the vaccination coverage of VarV-1 remained almost the same.

Therefore, Suzhou officially included VarV-1 in EPI on 1 April 2018. The curve rose at a slower rate in 2018–2019. The incidence of varicella in Suzhou demonstrated a yearly decreasing trend with the inclusion of VarV-2 in EPI on 1 January 2020 in Suzhou.

Some studies [10,11,12] point to a post-age shift after varicella vaccination. In this study, the median age of varicella cases in Suzhou shifted from 7.08 to 9.68 in 2021 after including the varicella vaccine in the EPI starting in 2018. However, the relationship between onset age of varicella cases and vaccination in a longer period needs further observation.

## 4. Discussion

Suzhou is one of the earliest cities to opt for varicella inclusion in passive surveillance and incorporate the varicella vaccine in the local EPI. This study demonstrates that the overall estimated incidence of varicella has substantially decreased in Suzhou, Jiangsu, following the implementation of the two-dose varicella vaccination programme for children in 2020. The annual average reported incidence was 126.19 per 100,000 in Suzhou, which was higher than that in other regions [5,11]. Possible reasons include the high sensitivity of local surveillance for varicella in Suzhou.

The comparison of the changes in regional incidence rates in the areas of Suzhou from 2012 to 2021 found that a high incidence rate was associated with population density and susceptibility. For example, the incidences were more elevated in Changshu, SIP, and SND, all districts with a large external population. The migratory resident population in each district accounted for 44.73%, 55.88%, and 57.59% of the total local population. Changshu, SIP, and SND had booster varicella vaccination rates of 0.04%, 0.65%, and 0.24%, respectively, throughout the three sites before the inclusion of varicella vaccine in the EPI in Suzhou.

The leading varicella incidence group had children aged 1–6 years and the children in this population segment were primarily in kindergarten. This group is the target population for Suzhou’s routine varicella vaccination programme, which may lead to an annual decrease in varicella incidence for this age group. Notably, following the implementation of the varicella vaccination schedule for the target population, a fluctuation in the incidence for children aged 0–1 years was reported. There are three rational hypotheses for this anomaly: first, the varicella vaccination rate for children aged 1 to 6 years increased significantly after the inclusion of varicella vaccination into the local EPI, potentially reducing the natural immunisation of children aged 0–1 years. Second, the declined varicella incidence across the whole population can cause a decreased level of varicella antibodies in mothers, which increases the susceptibility of children aged 0–1 years [13,14,15]. Finally, the level of diagnosis of varicella cases in medical institutions and reporting rates increased, perhaps more realistically representing the actual incidence data.

An upward shift in the infection age should be addressed because adult patients with varicella have more severe symptoms and a higher risk of death and complications than children [16,17,18]. However, varicella vaccination is still recommended school-aged children to reduce the risk of infection risk [19,20,21].

There is a clear seasonal pattern of varicella incidence, with a sizeable annual peak between October to January and a small peak from April to July, which is generally consistent with previous studies [7,20,22]. The presence of this epidemic feature may be attributed to the temperate region of Suzhou. A related study noted that the transmission of VZV does not require intimate contact; moreover, geo-climatic factors may limit its transmission characteristics. It has been documented [23] that UV radiation is the most remarkable difference between tropical and temperate regions. The presence of UV light reduces the rupture of the vesicles of inactivated viruses in infectious varicella cases before or after. This activity would reduce the transmissibility of varicella in the tropics and explain why the peak incidence of varicella in temperate regions is in winter and spring. Meanwhile, total rainfall, barometric pressure, and sunshine hours were positively correlated with varicella incidence [24]. The results suggest that preventing and controlling varicella infections during winter is crucial. In winter, because of the lower outdoor temperature, people choose more activities indoors, with more inter-individual contact and air conditioning for heating, which may contribute to the higher varicella cases.

Recent studies shows that the onset and vaccination of varicella may affect other disease, including cardiovascular disease and systemic lupus erythematosus (SLE). Patients with cardiovascular disease and SLE are more likely to develop a severe form of VZV. For instance, VZV causes cardiovascular disease, such as heart failure, ischemic stroke, and myocardial infarction. Therefore, vaccination may be proactively requested for patients with cardiovascular disease [25]. More clinical studies are needed to determine which cardiovascular diseases would benefit most from this vaccination and the impact of vaccination on primary prevention of cardiovascular disease in healthy populations. In patients with SLE, VZV infection within the last 3 months is associated with an increased risk of disease flares [26]. The relationship between varicella and microscopic matter can be discussed in a follow-up study [27,28].

Notably, the incidence of varicella, the number of reported cases and public health emergency events in Suzhou in 2020 decreased because of the COVID-19 epidemic. This phenomenon is not an exception and has been observed worldwide in different countries and cites [29,30]. In addition, diseases with a similar mode of transmission to varicella, such as hand, foot, and mouth disease, present varying degrees of decline in reported incidence in 2020 [31]. In 2020, the National Health Commission proposed the “six dos and six don’ts”, such as washing hands regularly, not eating without washing hands, sharing meals, and avoiding gathering for meals. The general public has subsequently developed a heightened awareness of personal hygiene, resulting in fewer public contacts and less exposure to infectious aerosols. These initiatives have interrupted the traditional transmission routes of respiratory infections [32]. This result suggests that reducing crowd gathering, frequently wearing masks, environmental disinfection, and the public’s focus on personal hygiene contribute to reducing the spread of varicella disease.

The varicella vaccine, an effective means of preventing varicella disease, did not become available in China until the advent of domestic and imported vaccines in 2000. Today, after years of process innovation and quality improvement, the Chinese varicella vaccine has better quality, efficacy, safety, and reliability. The vaccine effectively protected children in studies related to the domestic VarV-1 [33]. During surveillance, the efficacy of the varicella vaccine was 87.1% (95% confidence interval (CI), 69.7–94.5). However, VarV-1 could not interrupt the transmission of varicella compared to VarV-2. VarV-2 vaccination has become the preferred choice because of the high breakthrough rate of VarV-1 [34]. The evaluation of the immunogenicity and safety of live attenuated VarV-2 manufactured by China vaccine corporation revealed that varicella virus has good immune persistence for one to three years after initial immunisation. Vaccine boosters in children aged one to three years of age after the basic immunisation restore the specific immune response to VZV without increased safety concerns [21].

Relevant studies found that the estimated efficacies of VarV-1 and VarV-2 vaccinations were 94.4% and 98.3% (*p* < 0.001), respectively, and the measurable serum antibodies persisted for nine years in all the subjects [35]. The matched odds ratio for VarV-2 vs. VarV-1 was 0.053, and the effectiveness of VarV-2 was as high as 98.3% [36]. This outcome is generally consistent with findings obtained in China.

In 2014, the World Health Organisation recommended establishing a disease surveillance system to assess the disease burden of varicella and that the varicella vaccine could be added into routine childhood immunisation programmes when national resources are adequate and when 80% or more vaccine coverage is achieved [16]. However, until now, the varicella vaccine was not included in the China immunisation programme but is only administered as a self-paying vaccine [22]. A scientific, rational, comprehensive, and systematic indicator system for prioritising candidate vaccines to consider the inclusion in the EPI was recently effectively established for Chinese experts using a modified Delphi technique. The varicella vaccine was in the top position with the highest score (6.91) [37]. This outcome indicates that including the varicella vaccine in the national immunisation programme is necessary.

The varicella disease burden was reduced in areas where the varicella vaccine was administered free of charge in a disease burden-related study [38,39]. In this study, the incidence of varicella vaccine exhibited a declining trend year by year after implementing the complete free vaccination in Suzhou. The disease burden was positively correlated with the incidence that the population benefited from implementing this policy.

In conclusion, implementing the varicella vaccine has led to a decline in varicella incidence in Suzhou, especially among the target population. However, noteworthy changes were observed in the epidemiology characteristics after implementation. The median age of varicella infection has shifted to teenagers, and varicella incidence for one-year-old children has increased. The varicella incidence in Suzhou between 2012 and 2021 presents significant seasonal variations. Highlights from the study include acknowledging critical seasonal periods, areas, and populations. Morning checks and varicella prevention information must be developed in collective units where children tend to congregate. Importantly, a varicella surveillance system evaluation should be undertaken when necessary.

## Figures and Tables

**Figure 1 vaccines-10-01745-f001:**
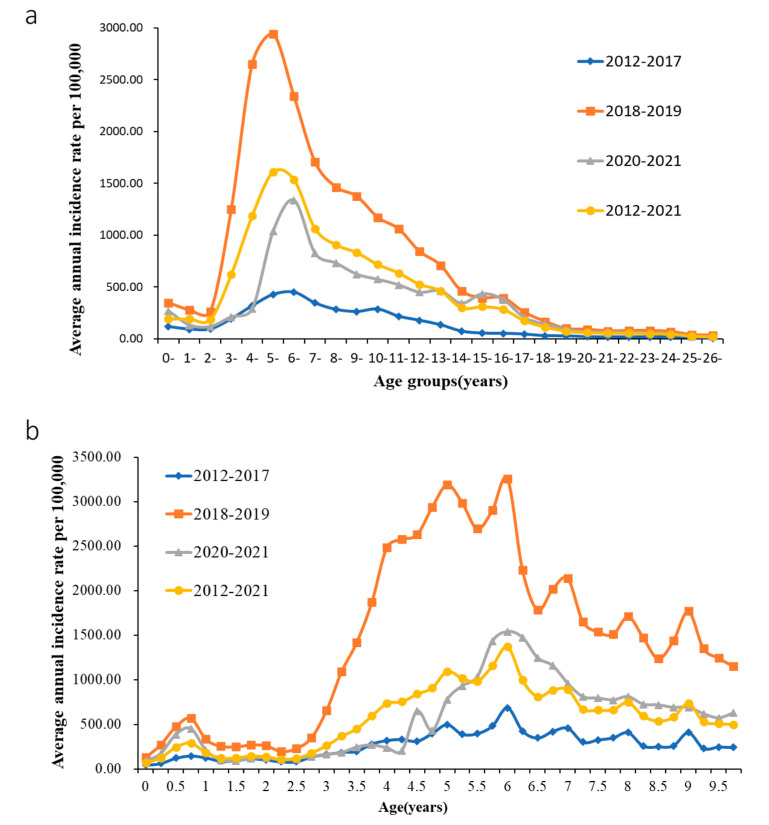
The age distribution of varicella incidences in Suzhou, 2016–2021. (**a**) Incidence of varicella in the whole population. (**b**) Varicella incidence among children aged < 10 years.

**Figure 2 vaccines-10-01745-f002:**
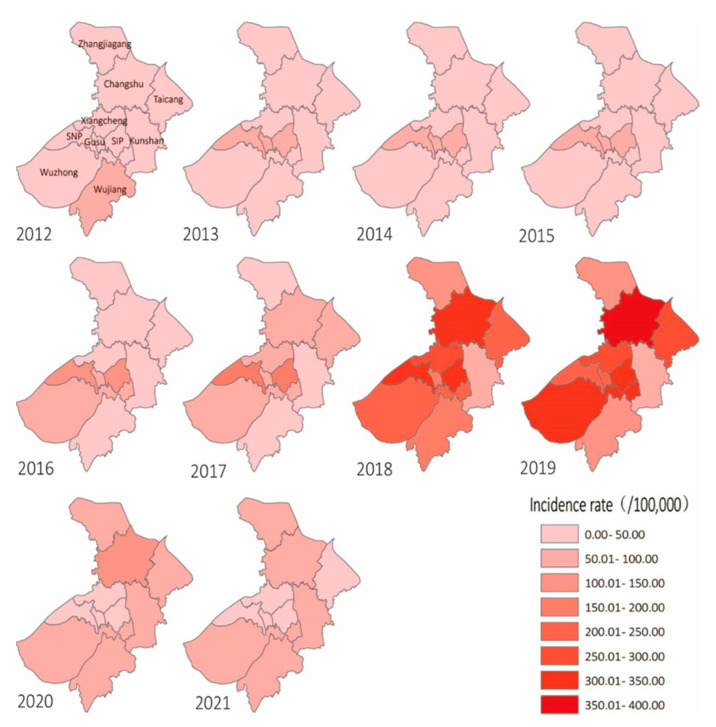
The district distribution of incidence rates in Suzhou, 2012–2021.

**Figure 3 vaccines-10-01745-f003:**
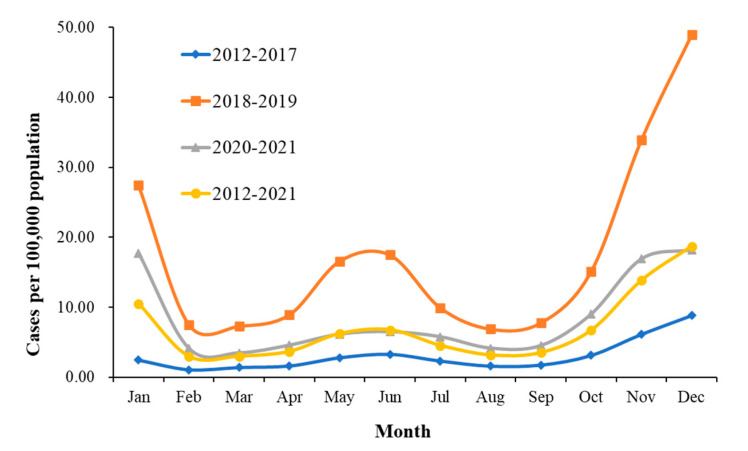
Incidence of varicella by month in Suzhou, 2012–2021.

**Figure 4 vaccines-10-01745-f004:**
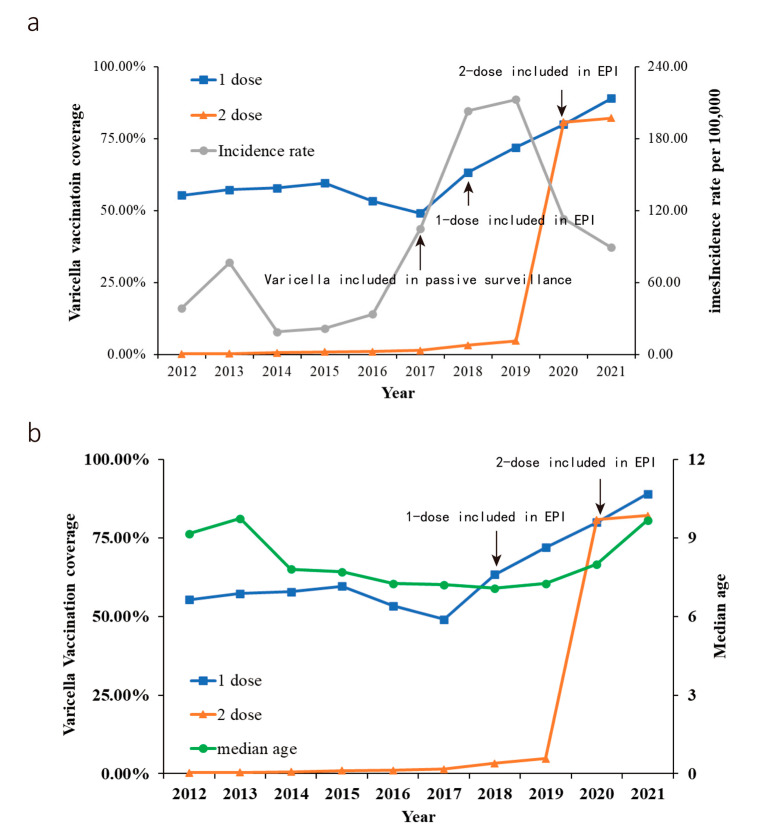
Evaluation of the relationship between vaccine vaccination coverage on varicella incidence or median age in Suzhou, Jiangsu, 2012–2021. (**a**) Change in varicella vaccination coverage and age-specific incidence. (**b**) Change in varicella vaccination coverage and median age.

**Table 1 vaccines-10-01745-t001:** The incidence rates and vaccination schedule of varicella in Suzhou, China, 2012–2021.

Period	Average Incidence (per 100,000)	χ^2^	*p*-Value	Vaccination Schedule
2012–2017	49.17	98.077	0.00	before the government implemented free varicella vaccination
2018–2019	207.98	36.83	0.00	1-dose varicella vaccine included in local EPI
2020–2021	101.26			2-dose varicella vaccine included in local EPI

*p*-value of χ^2^ test: the average of incidence compared with the next period.

**Table 2 vaccines-10-01745-t002:** The incidence rates of varicella by district in Suzhou, China, 2012–2021.

District	Average Incidence (per 100,000)				$\chi {}_{1}^{2}$	P_1_	$\chi {}_{2}^{2}$	P_2_
	2012–2021	2012–2017	2018–2019	2020–2021				
Gusu	80.51	22.05	145.83	81.86	91.265	0.00	17.973	0.00
Xiangcheng	74.36	30.30	76.67	54.49	20.101	0.00	3.7508	0.05278
Zhangjiagang	45.67	25.96	257.02	36.90	188.67	0.00	164.85	0.00
Kunshan	44.41	56.57	201.83	30.98	127.47	0.00	164.85	0.00
Taicang	74.51	21.46	249.96	58.19	192.37	0.00	119.34	0.00
SNP	117.01	77.07	316.98	43.21	146.06	0.00	208.08	0.00
Wujiang	58.77	10.37	141.01	56.25	112.74	0.00	36.42	0.00
Wuzhong	80.00	89.17	280.90	36.64	99.334	0.00	187.89	0.00
SIP	118.28	35.60	360.95	98.67	266.93	0.00	149.67	0.00
Changshu	113.29	44.81	211.47	54.12	108.37	0.00	93.233	0.00

P_1_ of $\chi {}_{1}^{2}$ test: comparison of the average annual incidence rate in 2012–2017 with the average annual incidence rate in 2018–2019; P_2_ of $\chi {}_{2}^{2}$ test: comparison of the average annual incidence rate in 2018–2019 with the average annual incidence rate in 2020–2021.

## Data Availability

Not applicable.

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
