# Peer review of "Epidemiological Characteristics of Varicella under Different Immunisation Strategies in Suzhou Prefecture, Jiangsu Province"

_vaccines, 2022, doi:10.3390/vaccines10101745_

Round 1

Reviewer 1 Report

Q1. Add in methods the study model design.

Q2. Figure 4 a. The grey line is not described in figure 4a.

Q3. In discussion paragraph you should focus on two aspects of varicella vaccination: the safety profile (and so the ratio risk/benefits) and the long-term immunogenicity of the vaccine. Please revise

Author Response

Dear Camellia Wang and Reviewers,

Thanks very much for taking your time to review this manuscript. We really appreciated all your comments and suggestions, which greatly enhanced the quality of our manuscript! Please find our itemized responses in below.

Thanks again!

Responds to the reviewers’ comment:

Reviewer #1

Point 1: Add in methods the study model design.

Response 1: Thank you for your reminder, and the information has been updated. In this study, we used descriptive statistics to analyze and process the relevant data. Please see line 97-108 for details.

Line 97-108 - Total and age-specific incidence rates were calculated based on the number of varicella cases and the study population from 1 January 2012 to 31 December 2021. The information on varicella cases was retrieved from the CISDCP, and the number of cases in each area of Suzhou was recorded according to the address information therein. The incidence rate of each area was then calculated.

Microsoft Office Excel (version 2016) and R for Windows (version 4.1.3) were used for data processing and statistical analyses. Mapping of districts used ArcGIS (version 10.8). Descriptive statistics were applied to describe the incidence in terms of person, place and time. Chi-squared or Pearson chi-squared tests were used to compare the incidences rates and vaccine coverage. The main focus was on changes in morbidity and vaccine coverage before and after the inclusion of the varicella vaccine in EPI and before and after the inclusion of VarV-1 and VarV-2. The level of statistical significance was set at P < 0.05.

Point 2: Figure 4.a. The grey line is not described in figure 4a.

Response 2: We are sorry for this minor error, and the information of grey line in figure 4a has been updated. The grey line in figure 4a represents the change in the incidence of varicella in Suzhou from 2012-2021.

Point 3: In discussion paragraph, you should focus on two aspects of varicella vaccination, the safety profile (and so the ratio risk/benefits) and the long-term immunogenicity of the vaccine. Please revise.

Response 3: It is really a good idea as reviewer suggested, and we have changed them all to meet reviewer’s thought as shown in following paragraphs.

In lines 299-308, we describe the good vaccination effectiveness, reliability and safety for both the one-dose varicella vaccine and two-dose varicella vaccine. Compared with 1-dose varicella vaccine, two-dose varicella vaccine had better protection.

In lines 309-313, we elucidated that varicella vaccine has good long-term immunogenicity.

In lines 325-329, we mentioned the benefit of reducing the disease burden by making varicella vaccination free of charge. Adopting a strategy of free varicella vaccination can lead to a decrease in the incidence of varicella, thus leading to a reduction in the varicella disease burden.

Thank you again and all the reviewers for the kind advice again. If you have any question, please contact us without hesitate.

Yours sincerely,

Lin Luan

Reviewer 2 Report

 The presented work "Epidemiological characteristics of varicella under the different immunisation strategies in Suzhou Prefecture, Jiangsu Province" by Zhuoyu Zhang, Yunyan Zhang, Na Liu, Jun Zhang, Jiaqi Xiao, Wenyu Wang, Juan Xu, Tianyu Wang, Lin Luan concerns the effects of vaccination on frequency of VZV infections in children in the Suzhou region. This subject fits well with the scope of Vaccines Special Issue: "Epidemiology, Vaccination and Public Health". However, in my opinion, the quality of the presentation is not sufficient for publication in its current form.

The main objections concern the presentation of the results. It is unclear, the figures are imprecise and their description is incomplete. This makes it very difficult to understand the analyzes. The discussion explains the results a bit. In my opinion, the work needs to be revised by a native speaker.

Specific Notes:

line 23 - The average annual reported incidence between districts is ... unclear

line 68 - Before 2018 April 1st, parents paid for their children's vaccination as voluntary self-paying - but line 39 -until now, the Varicella vaccine was not included in the China immunisation program but is only administered as a self-paying vaccine

line 88 - A total of 118,031 cases of varicella disease (including one fatal case) were reported in Suzhou - during what period?

line 95 - On 31 December 2021, there were 118,031 cases of varicella reported in Suzhou, where 19.07% of the study population was <4 years old; ... - please correct

line 119-126- please correct, unclear

Figure 3 - description from line 139 does not correspond to the figure

line 156 - there is no figure 5, it should be 4

fig 4 - a - which means gray line

blue line over 50% before the first dose is introduced (maybe before the first dose is refunded)

fig 4b - median age of vaccinated or incidents? Please, explain what this figure shows?

Author Response

Dear Camellia Wang and Reviewers,

Thanks very much for taking your time to review this manuscript. We really appreciated all your comments and suggestions, which greatly enhanced the quality of our manuscript! Please find our itemized responses in below.

Thanks again!

Responds to the reviewers’ comment:

Reviewer #2

We apologize for the poor language of our manuscript. The language and readability have been enhanced by the helps of native English speakers. We really hope that the flow and language level have been substantially improved.

Point 1: line 23 - The average annual reported incidence between districts is ... unclear

Response 1: Thank you for your reminder. Our original statement was ambiguous, and we corrected the information in the revised manuscript as shown in Line 28-29.

Line 28-29 - The average annual reported incidence between districts with a statistically significant difference (c2=120.36, p<0. 05).

Point 2: line 68 - Before 2018 April 1st, parents paid for their children's vaccination as voluntary self-paying - but line 39 -until now, the Varicella vaccine was not included in the China immunisation program but is only administered as a self-paying vaccine

Response 2: We are very sorry for the confusing information. The first sentence was intended to convey the situation of residents of Suzhou, while the second sentence was intended to clarify that varicella has not been included as a free vaccination in China. The reality is the varicella vaccination is not free throughout China, but only in localized areas (e.g., Suzhou). We have revised this section for this submission.

Point 3: line 88 - A total of 118,031 cases of varicella disease (including one fatal case) were reported in Suzhou - during what period?

Response 3: Thanks for the reminder. The period information has been updated. It is 2012 to 2021.

Line 109-110 – From 2012 to 2021, a total of 118,031 cases of varicella disease were documented in Suzhou, and the average annual reported incidence was 91.35 per 100,000.

Point 4: line 95 - On 31 December 2021, there were 118,031 cases of varicella reported in Suzou, where 19.07% of the study population was <4 years old; ... - please correct

Response 4: We apologize for the confusing information regarding the number of cases of varicella in each age group. The information has been modified as shown in line 119-121.

Line 119-121 - During 2012-2021, 18.00% of cases were reported in ≤4 years old, 40.11% of patients were aged between 5 and 9 years old, and 13.56%, 12.06% and 16.28% of the total cases belonged to age groups of 10-14, 15-19 and 20 years and above, respectively.

Point 5: line 119-126- please correct, unclear

Response 5: We are sorry for the confusing information. We have revised the paragraph accordingly. Please see line 151-161 for details.

Line 151-161 –

Table 2 shows the average annual incidence rates for each district in Suzhou from 2012 to 2021. The differences in the average annual incidence rates between the districts were statistically significant (c2 = 85.482, p < 0.05). Table 2 shows that the differences in incidence in Suzhou were statistically significant before and after the inclusion of the varicella vaccine in EPI and compared to the incidence by region in 2018-2019 and 2020-2021.

Figure 2 illustrates that the color deepened in some area in 2017 compared to 2012-2016.Though the color showed a continuous process of deepening in all the area of Suzhou in 2018 and 2019, the color became lighter again in 2020-2021.

From 2012 to 2021, the top three average annual incidence in Suzhou occurred in Suzhou Industrial Park (SIP) (118.28/100,000), Changshu (113.29/100,000) and Suzhou national Hi-tech District (SND) (117.01/100,000). The highest incidences in 2012, 2013, 2014-2017 and 2018-2021 were occurred in Wuzhong (69.95/100,000), SIP (63.05/100,000), SND (150.10/100,000) and Changshu (229.81/100,000), respectively.

Point 6: Figure 3 - description from line 139 does not correspond to the figure

Response 6: Thank you for the reminder. We have corrected the errors as shown in line 179-181.

Line 179-181 – The average monthly incidence before the varicella vaccine implementation was 3.04 per 100,000, which increase to 17.33 per 100,000 in 2018-2019 and decrease to 8.44 per 100,000 after the VarV-2 implementation.

Point 7: line 156 - there is no figure 5, it should be 4

Response 7: We are very sorry for our negligence of the reference format. We have revised the manuscript in this submission.

Point 8: fig 4 - a - which means gray line, blue line over 50% before the first dose is introduced (maybe before the first dose is refunded)

Response 8: We are very sorry for our incorrect writing and it is rectified at grey line in figure 4a. The grey line in figure 4a represents the change in the incidence of varicella in Suzhou from 2012-2021 and we analyze it in lines 200-207.

Line 200-207 - Figure 4 (a) shows the incidence of varicella by year in Suzhou from 2012 to 2021. A slight fluctuation in the incidence of varicella was observed from 2012 to 2014. This change may be attributed to other factors, thus making health workers more motivated to report varicella. Following the inclusion of varicella in passive surveillance on 1 June, 2017, varicella incidence rose sharply from 2017 to 2018, suggesting an underestimation of previous varicella incidence. From 2012 to 2017, VarV-1 did not show a notable downward trend in varicella incidence; however, the vaccination coverage of VarV-1 remained almost the same.

Point 9: fig 4b - median age of vaccinated or incidents? Please, explain what this figure shows?

Response 9: The median age in Figure 4 (b) is stated for the incidents. For the detailed description of Figure 4(b) see line 211-214.

Line 211-214 - In this study, the median age of varicella cases in Suzhou shifted from 7.05 to 9.68 in 2021 after including the varicella vaccine in the EPI starting in 2018. However, the relationship between onset age of varicella cases and vaccination in a longer period needs further observation.

Thank you again and all the reviewers for the kind advice again. If you have any question, please contact us without hesitate.

Yours sincerely,

Lin Luan

Reviewer 3 Report

This is an interesting and meaningful study, I would like to recommend this manuscript for publication after minor revision:

1.     Do other diseases affect the onset and vaccination of varicella, such as inflammation, tumor, intestinal disease < Xiaojing Sun, Zhonghua Xue, Aqeela Yasin, et al., Colorectal Cancer and Adjacent Normal Mucosa Differ in Apoptotic and Inflammatory Protein Expression, Engineered Regeneration 2 (2022) 279-287.> or other metabolic diseases or chronic diseases < Xueqi Zhang, Yachen Hou, et al. The role of astaxanthin on chronic diseases. Crystals 2021; 11: 505.>? The authors could briefly mention in the Introduction.

2.     In the main text, the cited ref. number should keep a space between the previous word.

3.     The reference format is not consistent with the journal requirement. Please check and revise it.

4.     The manuscript need further discussion, not only display the data.

Author Response

Dear Camellia Wang and Reviewers,

Thanks very much for taking your time to review this manuscript. We really appreciated all your comments and suggestions, which greatly enhanced the quality of our manuscript! Please find our itemized responses in below.

Thanks again!

Responds to the reviewers’ comment:

Reviewer #3

Point 1: Do other diseases affect the onset and vaccination of varicella, such as inflammation, tumor, intestinal disease < Xiaojing Sun, Zhonghua Xue, Aqeela Yasin, et al., Differ in Apoptotic and Inflammatory Protein Expression, Engineered Regeneration 2 (2022 Colorectal Cancer and Adjacent Normal Mucosa) 279-287.> or other metabolic diseases or chronic diseases < Xueqi Zhang, Yachen Hou, et al. The role of astaxanthin on chronic diseases. Crystals 2021; 11: 505.>? The authors could briefly mention in the Introduction.

Response 1: According to the Reviewer’s suggestion, we have added relevant content and added these two papers. Please see line 270-280 for details.

Recent studies shows that the onset and vaccination of varicella may affect other disease, including cardiovascular disease and systemic lupus erythematosus (SLE). Patients with cardiovascular disease and SLE are more likely to develop a severe form of VZV. For instance, VZV causes cardiovascular disease, such as heart failure, ischemic stroke and myocardial infarction. Therefore, vaccination may be proactively requested for patients with cardiovascular disease [25]. More clinical researches are needed to determine which cardiovascular diseases would benefit most from this vaccination and the impact of vaccination on primary prevention of cardiovascular disease in healthy populations. In patients with SLE, VZV infection within the last 3 months is associated with an increased risk of disease flares [26]. The relationship between varicella and microscopic matter can be discussed in a follow-up study [27,28].

  1. Cersosimo, A.; Riccardi, M.; Amore, L.; Cimino, G.; Arabia, G.; Metra, M.; Vizzardi, E. Varicella Zoster Virus and Cardiovascular Diseases. Monaldi Arch Chest Dis 2022.
  2. Sun, F.F.; Chen, Y.; Wu, W.L.; Guo, L.; Xu, W.W.; Chen, J.; Sun, S.H.; Li, J.J.; Chen, Z.W.; Gu, L.Y.; et al. Varicella Zoster Virus Infections Increase the Risk of Disease Flares in Patients with SLE: A Matched Cohort Study. Lupus Sci Med 2019, 6, e000339.
  3. Zhang, X.; Hou, Y.; Li, J.; Wang, J. The Role of Astaxanthin on Chronic Diseases. Crystals 2021, 11, 505.
  4. Sun, X.; Xue, Z.; Yasin, A.; He, Y.; Chai, Y.; Li, J.; Zhang, K. Colorectal Cancer and Adjacent Normal Mucosa Differ in Apoptotic and Inflammatory Protein Expression. Engineered Regeneration 2021, 2, 279–287.

Point 2: In the main text, the cited ref. number should keep a space between the previous word.

Response 2: We are very sorry for our negligence of the cited ref. number. We have revised the manuscript in this submission.

Point 3: The reference format is not consistent with the journal requirement. Please check and revise it.

Response 3: We are very sorry for our negligence of the reference format. We have revised the manuscript in this submission.

Point 4: The manuscript need further discussion, not only display the data.

Response 4: It is really a good idea as reviewer suggested, and we made the following changes based on your suggestion.

  1. We analyzed the regional incidence of varicella in Suzhou from 2012-2021 exclusively with the population mobility and full vaccination rate in the area (line 233-240).
  2. We address the possible reasons for the increased incidence of varicella in children within one year of age after the inclusion of varicella in EPI (line 246-253).
  3. We have included in the discussion section the influence of environmental factors on varicella disease in line 260-271.
  4. We added two aspects of varicella vaccination, the safety profile (and so the ratio risk) and the long-term immunogenicity of the vaccine in line 296-313.
  5. Based on the Chinese context and recent research findings, we present our view on the inclusion of varicella vaccine in China (line 317-324).
  6. We have added a section on the disease burden of varicella (line 325-329).

Thank you again and all the reviewers for the kind advice again. If you have any question, please contact us without hesitate.

Yours sincerely,

Lin Luan

Round 2

Reviewer 2 Report

In my opinion, the manuscript meets the requirements for publication in the Vaccines journal.

Author Response

Dear Camellia Wang and Reviewers,

Thanks very much for taking your time to review this manuscript. We really appreciated all your comments and suggestions, which greatly enhanced the quality of our manuscript!

Thanks again!

Yours sincerely,

Lin Luan